# Halorotetin A: A Novel Terpenoid Compound Isolated from Ascidian *Halocynthia rotetzi* Exhibits the Inhibition Activity on Tumor Cell Proliferation

**DOI:** 10.3390/md21010051

**Published:** 2023-01-12

**Authors:** Jianhui Li, Shanhao Han, Yuting Zhu, Bo Dong

**Affiliations:** 1Fang Zongxi Center, MoE Key Laboratory of Marine Genetics and Breeding, College of Marine Life Sciences, Ocean University of China, Qingdao 266003, China; 2Laoshan Laboratory, Qingdao 266237, China; 3Institute of Evolution & Marine Biodiversity, Ocean University of China, Qingdao 266003, China

**Keywords:** ascidian, *Halocynthia roretzi*, terpenoid, antitumor, hepatocellular carcinoma

## Abstract

*Halocynthia roretzi*, the edible ascidian, has been demonstrated to be an important source of bioactive natural metabolites. Here, we reported a novel terpenoid compound named Halorotetin A that was isolated from tunic ethanol extract of *H. roretzi* by silica gel column chromatography, preparative layer chromatography (PLC), and semipreparative-HPLC. ^1^H and ^13^C NMRs, ^1^H-^1^H COSY, HSQC, HMBC, NOESY, and HRESIMS profiles revealed that Halorotetin A was a novel terpenoid compound with antitumor potentials. We therefore treated the culture cells with Halorotetin A and found that it significantly inhibited the proliferation of a series of tumor cells by exerting cytotoxicity, especially for the liver carcinoma cell line (HepG-2 cells). Further studies revealed that Halorotetin A affected the expression of several genes associated with the development of hepatocellular carcinoma (HCC), including oncogenes (*c-myc* and *c-met*) and HCC suppressor genes (*TP53* and *KEAP1*). In addition, we compared the cytotoxicities of Halorotetin A and doxorubicin on HepG-2 cells. To our surprise, the cytotoxicities of Halorotetin A and doxorubicin on HepG-2 cells were similar at the same concentration and Halorotetin A did not significantly reduce the viability of the normal cells. Thus, our study identified a novel compound that significantly inhibited the proliferation of tumor cells, which provided the basis for the discovery of leading compounds for antitumor drugs.

## 1. Introduction

Malignant tumors seriously endanger human health. In recent years, cancer incidence and death rates have been on the rise around the world [1]. The search for effective and little cytotoxicity drugs is a promising strategy for cancer therapy. Natural products are a major source of drugs, and nearly 70% of FDA-approved anticancer drugs are derived from natural products or their derivatives [2]. Marine invertebrates and their associated bacterial symbionts represent organisms with great potential for the discovery of novel bioactive natural products [3]. For instance, Discodermolide from the marine sponge inhibits the binding of paclitaxel to tubulin polymers, enhances tubulin nucleation reactions more potently than paclitaxel, and inhibits the growth of paclitaxel-resistant cells. It has excellent potential to be applied in clinical treatment [4]. Moreover, marine bacteria [5], fungi [6], and coral [7] are also important sources of bioactive natural products.

As a tunicate, ascidian locates the evolutionary node from invertebrate to vertebrate [8]. The diversity and wide adaptability of ascidians determine the novel structure and diverse functions of bioactive natural products derived from ascidians. Currently, there are a variety of natural anticancer products from ascidians [9]. The most popular ascidian-derived natural anticancer product is Ecteinascidin-743 (ET-743). ET-743 is an alkaloid compound derived from *Ecteinascidia turbinate* [10] and is cytotoxic to a variety of cancer cells [11]. ET-743 is a new class of tumor DNA binding agents with a unique mechanism of action, which can inhibit transcription-dependent nucleotide removal repair pathways in vitro and promote p53 pathway-independent apoptosis by affecting the cell cycle [12]. In addition, ET-743 also shows high application value in clinical treatment. It has significant cytotoxicity to soft tissue sarcoma, which is one of the most difficult tumors to treat. The survival rate of patients treated with ET-743 has been significantly improved [13]. Moreover, many bioactive peptides with antitumor activity are also isolated from ascidians. Vitilevuamide is a dicyclic peptide isolated from *Didemnum cuculiferum* and *Polysyncranton lithostrotumz*. It can inhibit microtubule polymerization and lead to cell cycle arrest, then producing cytotoxicity. In mice, Vitilevuamide relieved the symptoms of leukemia and significantly improved survival time. Because of its outstanding antitumor properties, it has been approved for preclinical trials [14]. Besides this, there are many bioactive natural products derived from ascidians, indicating that ascidians are an important source of bioactive natural products.

*Halocynthia roretzi* belongs to the phylum chordate, which is an edible ascidian [15]. The body of *H. roretzi* can be divided into three parts (tunic, stolon, and endocyst). The exterior is surrounded by a tunic. There is a stolon at the top of the tunic. Inside the tunic is body tissue, which is edible. The body tissues of *H. roretzi* are rich in nutrients, such as polyunsaturated fatty acids [16], carotenoids, and other micronutrients [17]. Many bioactive natural products produced by ascidians are derived from their symbionts. A previous report showed that *H. roretzi* has many symbiotic microorganisms [18]. Hence, we hypothesized that *H. roretzi* could also produce natural products with biological activities. There have been some previous reports about bioactive natural products derived from *H. roretzi*, such as erucamide [19] and some bioactive peptides [20]. Moreover, *H. roretzi* has also been reported to prevent diabetes [21]. However, there are few reports on the antitumor activity of natural small molecules in *H. roretzi* and previous reports have not gone beyond a relatively preliminary stage. Therefore, we use *H. roretzi* as material and try to find the antitumor natural products. 

## 2. Results

### 2.1. Isolation and Purification of Halorotetin A

The tunic, stolon, and endocyst are three parts of *H. roretzi* (Figure 1A). The tunic was dissected, crushed into small granules, and soaked in 95% ethanol for one week. Tunic extract (TE) was then examined for the cytotoxicity of tumor cells. The liver carcinoma cell line (HepG-2 cells) was used for toxicity detection. As shown in Figure 1B, the TE had significant cytotoxicity. Then, the extract was separated by silica gel column chromatography with different eluents based on the polarity. After separation by silica gel column chromatography, we obtained two parts named SGC-I and SGC-II. After cytotoxicity verification, the SGC-II had tumor cell cytotoxicity (Figure 1C and Appendix A). We further used preparative layer chromatography (PLC) to isolate SGC-II. The flow phase system used in PLC was methylene chloride: methanol = 30:1. After separation by PLC, the SGC-II was then divided into different parts (Appendix A). To identify which part had tumor cell cytotoxicity, we performed the cytotoxicity assay, as shown in Figure 1D and Appendix A, and the cytotoxic part PLC-VI was verified (Appendix A, the red circle). To further purify the active compounds, we chose HPLC for the subsequent purification. As shown in Appendix A, the active part was divided into several parts. To investigate which part had tumor cell cytotoxicity, we performed cytotoxicity tests. The cytotoxic part of the tumor cells is circled in Appendix A and named Halorotetin A (Figure 1E). 

### 2.2. Structural Identification of Halorotetin A

To identify the structure of the compound, we first performed HPLC to examine and verify the purity of the Halorotetin A (Figure 2A). This compound showed UV absorption maxima at 349 nm (Appendix A). The molecular formula of the compound was determined to be C_18_H_24_O_2_ by the HRESIMS data (Appendix A). The ^1^H NMR spectrum and HSQC spectrum of the compound showed that it contained five methyl groups, and the methyl group at 2.28 ppm connected to unsaturated carbon. There were also two groups of methylene signals in the high-field region. Their chemical inequivalence characteristics indicated that the two methylene groups were located in the cyclic structure. In addition, there was a hypomethyl hydrogen signal around 3.93 ppm and three aromatic hydrogen or alkene hydrogen signals in the low field region in the 6.00–8.00 ppm range. The ^13^C NMR spectrum showed that the compound had a total of 18 carbon atoms including 11 carbons linked to hydrogen and seven quaternary carbons, where a signal at 199.9 ppm indicated the presence of an aldehyde or carbonyl group in the compound. In the COSY spectrum, the correlation of H-2/3/4 suggested that the compounds contain -CH_2_-CH(OH)-CH_2_- structural fragments and combined with the HMBC correlation between CH_3_-15/CH_3_-16 and C-1/2/6 and the HMBC correlation signal between CH_3_-17 and C-4/6, the cyclohexane partial fragment contained in the structure was inferred. In addition, in the COSY spectrum, the correlation of H-10/11/12 indicated that it contained two adjacent double-bond structures, and the correlation between H-12 and C-10/13 in the HMBC spectrum and the correlation between H-14 and C-12/13 indicated that methyl C-14 was linked to an unsaturated carbonyl group and attached to the C-12 terminus. The correlation between H-10 and C-18 in the HMBC spectrum, and the HMBC correlation between H-18 and C-8/9, further complete the sidechain structure. In addition to the quaternary carbon of C-8, there was also a seasonal carbon signal. It was finally speculated that C-8/9 formed an alkyne group to connect the cyclohexane moiety to the side chain moiety. In the NOESY spectrum, CH_3_-18 was NOE related to H-11/14, and there was an NOE correlation between H-10 and H-12, so the double bond configuration of C-9/10 was an E configuration. In addition, we detected the optical rotation value and found that the optical rotation value was negative ([α]_D_ = −243.85). Meanwhile, we also detected the compound with a similar structure of the Halorotetin A. The optical rotation of previous compounds such as Halorotetin A was negative [22,23]. Hence, we thought that the absolute configuration of OH group at C-3′ was *R*-configuration. The infrared absorption (IR) indicated that the carbonyl (1585 cm^−1^), -C=C- (1709, 1662, 1585 cm^−1^), and -C≡C- (2200 cm^−1^) were present in the compounds (Appendix A). This evidence revealed that Halorotetin A is a novel terpenoid compound (Figure 2B and Appendix A, Table 1) isolated from *H. roretzi*, and the chemical name of Halorotetin A is (3E,5E)-8-(4-hydroxy-2,6,6-trimethylcyclohex-1-enyl)-6-methylocta-3,5-dien-7-yn-2-one.

### 2.3. Halorotetin A Inhibits the Proliferation of a Series of Tumor Cells

We investigated the inhibition of Halorotetin A on tumor cell proliferation using several tumor cell lines and a mouse fibroblast cell line (L929). The results showed that the Halorotetin A had cytotoxicity to all tumor cell lines after treatment for 48 h (Figure 3A). In addition, at the highest test concentration, the toxicity of the compound to most of the tested tumor cell lines did not exceed 50%, indicating that although the compound could inhibit the proliferation of different tumor cell lines, a higher dosage was required to achieve stronger cytotoxicity. Notably, the proliferation inhibition rate of the Halorotetin A on L929 was less than 20% at 80 μM concentration (Figure 3A), demonstrating that Halorotetin A was less toxic to normal cells. In addition, the semi-inhibitory concentrations (IC_50_s) of the Halorotetin A against four types of tumor cell lines were calculated. As shown in Figure 3B, the HepG-2 cell line was most sensitive (the lowest IC_50_) to Halorotetin A compared with other examined tumor cell lines. Subsequently, we performed time-gradient toxicity detection on HepG-2 cells and L929 cells and found that the toxicity of the Halorotetin A to HepG-2 cells but not to L929 became more evident with the extension of time (Figure 3C). 

### 2.4. Halorotetin A Shows the Cytotoxicity to HepG-2 Cells but Much Less to L929 Cells

To investigate whether Halorotetin A is as effective as previously developed clinical antitumor agents, we compared the cytotoxicity of Halorotetin A and doxorubicin on HepG-2 cells and L929 cells at the same concentration. As shown in Figure 4A, at the concentration of 16 μg/mL, the cytotoxicity of Halorotetin A on HepG-2 cells was similar as that of doxorubicin. Meanwhile, the Halorotetin A showed less cytotoxicity on L929 cells than doxorubicin. In addition, morphological features of L929 and HepG-2 cells were observed under light microscopy (Figure 4B). These results indicated that the Halorotetin A might be a preeminent leading compound for antitumor drugs.

### 2.5. Halorotetin A Inhibits c-myc and c-met but Promotes the TP53 Transcription Level in HepG-2 Cells 

To elucidate the potential molecular mechanisms of Halorotetin A on HepG-2 cell growth, we examined the transcription levels of several genes associated with the development of hepatocellular carcinoma (HCC), including oncogenes (*PIK3CA*, *c-myc*, and *c-met*) and HCC suppressor genes (*TP53*, *PTEN*, *KEAP1*, *ARID1A*, *ARID2*, and *AXIN*) [24] by qRT-PCR. The results showed that the transcription level of *c-myc* and *c-met* were significantly decreased. Furthermore, the transcription level of *TP53* and *KEAP1* were significantly increased after Halorotetin A treatment (Figure 5). These results indicated that the Halorotetin A inhibited the proliferation of HepG-2 cells via regulating the expression of oncogenes and the suppressor genes. 

## 3. Discussion

Natural products are an important source of antitumor drugs. Nearly 70% of antitumor drugs approved by the FDA are derived from natural products or their derivatives. In this study, we isolated and purified a terpenoid compound from ascidian *H. roretzi* tunic ethanol extract, identified its structure, and revealed its high inhibitory effects on tumor cell proliferation, especially for HepG-2 cells, possibly via the *c-myc*. *c-met* and *TP53* pathways. 

Terpenoids are one of the most abundant natural products and important sources of leading compounds in drug research [25]. Many of them have been found to have biological activities [26]. Glycyrrhizic acid and oleanolic acid are two terpenoids with antitumor activity to inhibit the proliferation of leukemia cells [27]. Meanwhile, glycyrrhizic acid also has therapeutic potential for treating prostate cancer due to its potent inhibitory effects on the proliferation of several prostate cancer cells [28]. In addition, oleanolic acid not only inhibits the proliferation of osteosarcoma cells [29] but also has a preeminent anti-inflammatory function [30]. The compound isolated from ascidian in this study has much lower IC_50_ values on tumor cells than glycyrrhizic (3 mg/mL). Meanwhile, we hypothesized that the Halorotetin A might have other biological activities such as glycyrrhizic acid and oleanolic acid.

The development of HCC is associated with many genes. *c-myc* is highly expressed in many tumors and implicated in promoting cell division [31]. A previous study identified a *c-myc* inhibitor KJ-Pyr-9 with potent inhibitory effects on the proliferation of tumor cells at the concentration of 20 μM in vivo and in vitro [32]. In the present study, we found that the expression of *c-myc* was significantly decreased in HCC cells after treatment of isolated and purified Halorotetin A. *c-met* is a hepatocyte growth factor with a tyrosine kinase activity [33], which is not only involved in cell signaling transduction and cytoskeletal arrangement but also related to the regulation of cell proliferation [34]. The *c-met* is the target of many commercial antitumor drugs, such as crizotinib [35] and tepotinib [36]. It is worth mentioning that the expression of *c-met* was significantly decreased after Halorotetin A treatment. *TP53* is also a classical tumor suppressor gene, which is associated with cell cycle arrest, cell apoptosis, aging, and DNA repair [37]. The expression of *TP53* is significantly increased after Halorotetin A treatment. This is consistent with our previous work in which the *TP53* gene provided by water extract from the endocyst of *H. roretzi* inhibited the proliferation of tumor cells [19].

So far, an increasing number of bioactive natural products derived from ascidians have been reported. Some natural products are endogenously produced by the ascidians during various metabolic processes, and others are produced by the symbionts of ascidians. Meridianins are indole alkaloids isolated from ascidian *Aplidium meridianum* [38]. Arenimycin is a polyketide isolated from *Salinispora Arenicola*, a subsidiary bacterium of *E. terbinate*, which can inhibit the proliferation of human colon cancer cell lines [39]. *H. roretzi* not only produces antitumor metabolites but also other bioactive metabolites, such as compounds with antibacterial and antifungal activities [40]. On the other hand, a large number of bacteria have been identified from the gut of *H. roretzi* [41]. Furthermore, many metabolites are identified from the symbiotic microorganisms [18]. The compound identified in this study could be produced by *H. roretzi* itself or by its symbiont. Identification of the source of this compound is interesting and important for the large-scale production and isolation of this effective compound in the future. In conclusion, our work identified a novel terpenoid antitumor compound from an edible ascidian, which has a preeminent-inhibitory effect on tumor cell proliferation, providing a promising basis for the discovery of leading compounds in the development of antitumor drugs.

## 4. Materials and Methods

### 4.1. Materials

Sea squirts (*H. roretzi*) were collected from the XUNSHAN group (Weihai, Shandong, China) and were dissected into the endocyst, tunic, and stolon. The tunics were ground into 1 cm × 1 cm pieces by a juicer (JoYoung, Jinan, China) and soaked in five times the volume of 95% ethanol (LIRCON, Dezhou, China) for one week. The tunic ethanol extract (TE) was concentrated by a rotary evaporator (EYELA, Tokyo, Japan) before storage at 4 °C. All organic reagents were purchased from Sinopharm Chemical Reagent Co., Ltd. (SCR, Shanghai, China). 

### 4.2. Preparation of SGC-II by Silica Gel Column Chromatography 

The silica gel column was assembled with 300–400 mesh silica gel (Appendix A), and 25 mL TE was added into 18 g 200–300 mesh silica gel and then added into the silica gel column. After the silica gel column was loaded, the column was first infiltrated with petroleum ether, and then the whole column was wetted by petroleum ether. After that, the gradient elution was performed by ethyl acetate: petroleum ether = 1:4 (200 mL ethyl acetate, 800 mL petroleum ether), 1:3 (100 mL ethyl acetate, 300 mL petroleum ether), 1:2 (100 mL ethyl acetate, 200 mL petroleum ether), and 100% methanol (500 mL). The solvent in the eluate was dried by the rotary evaporator, then dissolved by methanol.

### 4.3. Preparation of Halorotetin A by PLC and HPLC 

SGC-II was separated and purified by the PLC silica gel plate (Merck, Darmstadt, Germany) in the dichloromethane-methanol 15:1 solvent system (30 mL dichloromethane, 2 mL methanol). The different fractions separated by PLC were dissolved by methanol, respectively. The precipitates, such as silica gel, were removed by filtration, concentrated by the rotary evaporator, and then re-dissolved by adding methanol. The final separation and purification of PLC-VI were carried out by semi-preparative HPLC (HITACHI, Tokyo, Japan) using water (WAHAHA, Hangzhou, China) and acetonitrile (HPLC) as the flow phase. The proportion of acetonitrile gradually increased from 20% to 100% (Appendix A). The column was 250 × 10 nm, 120 A C 18 reversed-phase silica gel column (SIGREEN, Beijing, China). After evaporation and concentration of each HPLC eluent, ten components were obtained by dissolving with methanol (The Halorotetin A retention time: 26 min).

### 4.4. Structural Identification of Halorotetin A

NMR spectra was recorded by a JEOL JEM-ECP NMR spectrometer (600 MHz for ^1^H NMR and 150 MHz for ^13^C NMR). HRESIMS was tested on a Thermo MAT95XP high-resolution mass spectrometer. HRESIMS [M + Na]^+^
*m/z* 295.1667 (calcd for C_18_H_24_O_2_Na, 295.1669).

### 4.5. Cell Culture

Human cervical cancer HeLa cell, mouse fibroblast L929 cell, human thyroid cancer BHT101 cell, and human breast cancer Mcf-7 cells were obtained from the Shanghai Cell Bank (Shanghai, China). Human liver carcinoma HepG2, Sk-Hep-1, and Bel-7402 cells were obtained from Xiguang Chen laboratory at the Ocean University of China (Qingdao, China), Medical College of Qingdao University (Qingdao, China), and Baoqin Han laboratory at the Ocean University of China (Qingdao, China), respectively. The HepG-2 cells, HeLa cells, BHT-101 cells, Mcf-7 cells, Sk-Hep-1 cells, and Bel-7402 cells were cultured in Dulbecco‘s modified eagle’s medium (DMEM) (Biological Industries, Kibbutz Beit, Israel), supplemented with 15% fetal bovine serum (FBS) (Biological Industries, Israel) and 1% penicillin-streptomycin (Biological Industries, Israel). The L929 cells were cultured in DMEM supplemented with 10% FBS (Gibco, Invitrogen, Carlsbad, CA, USA) and 1% penicillin-streptomycin. All cell lines were cultured at 37 °C with 5% CO_2_ incubator (ThermoFisher, Waltham, MA, USA).

### 4.6. MTT Assay for Cell Viability 

Next, 3-(4,5-dimethylthiazol-2-yl)-2,5-diphenil tetrazolium bromide (MTT) (Sigma-Aldrich, St. Louis, MO, USA) was used to detect the cell viability with the treatment of all samples. MTT was prepared at 5 μg/mL concentration using Phosphate-buffered saline (PBS) (Biological Industries, Israel). When the medium was removed, 100 μL DMEM medium containing 10% MTT was added. After 4 h of incubation at 37 °C with 5% CO_2_ incubator, the DMEM medium containing MTT was removed and 100 μL DMSO (Solarbio, Beijing, China) was added into each well. After 5 min, the absorbance of each well at 490 nm was measured using a microplate reader (TECAN, Mannedorf, Switzerland) after 15 s of horizontal oscillation.

### 4.7. RNA Extraction from HepG-2 Cells

HepG-2 cells for RNA extraction were centrifuged in 1.5 mL RNase-free centrifuge tubes for several seconds using a low-speed centrifuge to remove the supernatant. Then, 1 mL RNAsio Plus (Takara, Beijing, China) was added to the centrifuge tube and then the cells were fully lysed by vortexing and shaking for 40 s. Then 200 μL chloroform was added to the centrifuge tube, followed by vortexing and shaking for 40 s until the liquid turned pink. Then the centrifuge tubes were put on ice for 10 min. Subsequently, 400 μL supernatant was taken into a new 1.5 mL RNase-free centrifuge tube after centrifugation at 12,000 rpm for 15 min at 4 °C using a refrigerated high-speed centrifuge. Then 400 μL isopropanol was added. Samples were centrifuged at 12,000 rpm for 15 min at 4 °C using a refrigerated high-speed centrifuge after leaving for 15 min at room temperature. The supernatant was removed, and 1 mL 75% ethanol (RNase-free) was added to the precipitate. Samples were centrifuged at 12,000 rpm for 15 min at 4 °C using a refrigerated high-speed centrifuge. The supernatant was removed, the residual ethanol was volatilized entirely at room temperature, and 20 μL DEPC-treated water (Biological Industries, Israel) was added to the centrifuge tube to dissolve the precipitated RNA of HepG-2 cells. The concentration and quality of RNA were detected by Nanodrop (Eppendorf, Hamburg, Germany) and 1% agarose gel electrophoresis (Appendix A). The RNA of HepG-2 cells was stored at −80 °C until performing the reverse transcription.

### 4.8. Reverse Transcription

The reverse transcription reaction mixture was mixed by R223-01 HiScript^®^ II Q RT SuperMix Kit (Vazyme, Cat. No.: Vazyme.R223-01, Nanjing, China), which was 20 μL including 4 μL 5 × qRT SuperMix II and 500 ng total RNA and DEPC-treated water configured. The mixture was treated at 50 °C for 15 min, followed by 85 °C for 5 s in a PCR apparatus (BioRaD, Hercules, CA, USA).

### 4.9. Detecting of Gene Expression

The cDNA obtained by reverse transcription was carried out qRT-PCR as a pre-experiment. According to the Ct values obtained in the pre-experiment, 2 μL cDNA was then taken, and the Ct value of cDNA was diluted to 20 μL with dd water as the optimal cDNA concentration required for the qRT-PCR reaction of different target genes. The software Primer Premier 5.0 was used to design qRT-PCR primers. The primers used in the experiment were listed in Appendix A. The qRT-PCR reaction mixture was mixed using a ChamQ SYBR Color qPCR Master Mix Kit (Vazyme, Q431-02, China), which was 20 μL including 10 μL 2× ChamQ SYBR Color qPCR Master Mix, 0.4 μL forward and reverse primer of the target gene, respectively, 8.2 μL DEPC-treated water and 1 μL cDNA. The pre-denaturation temperature of PCR amplification was 95 °C for 5 min, followed by 40 cycles (95 °C for 30 s, 59 °C for 30 s, and 72 °C for 30 s), and then the determination of the melting curve (95 °C for 15 s, 60 °C for 60 s, 95 °C for 15 s). The β-actin gene was used as a reference gene for the qRT-PCR.

### 4.10. Statistical Analysis

Statistical analyses were performed by GraphPad Prism version 9.0 for Mac (GraphPad Software, San Diego, CA, USA). For boxplot: the vertical line represents 1.5 IQR, the upper and lower hinges are presented as the 75% and 25% quantiles, respectively, and the center line means the median value. For column: the data were presented as mean ± standard deviation (SD). Significance was determined by one-way ANOV, *t*-test after Welch’s correction. All data were determined by normality tests. The relative cell viability was calculated as follows: the absorbance of sample-treated cells at 490 nm divided by the absorbance of methanol-treated cells at 490 nm after 48 h. Methanol-treated cells were used as control and normalized. The IC_50_ fitting curve was drawn as follows: nonlinear regression was performed after logarithmic transformation of the concentration. The formula used for calculating IC_50_: Y = Bottom + (Top × Bottom)/[1 + 10^(Log (IC_50_) × ^^X)^] *×* HillSlope, Top was manually set to 1 and The Bottom was manually set to 0, the fitting curve of IC_50_ obtained X and Y. The results of qRT-PCR were calculated by the 2^−ΔΔCT^ method, and the control group was normalized.

## 5. Conclusions

The structure of a novel terpenoid Halorotetin A, isolated from the sea squirt *H. roretzi,* was determined to be (3E,5E)-8-(4-hydroxy-2,6,6-trimethylcyclohex-1-enyl)-6-methylocta-3,5-dien-7-yn-2-one by HRESIMS, IR, and NMR spectral data. The Halororetin A inhibited a series of tumor cell proliferation, especially for HepG-2 cells via affecting the expression of HCC development-associated genes. These findings provide the basis for the discovery of leading compounds for antitumor drugs from sea squirt. 

## Figures and Tables

**Figure 1 marinedrugs-21-00051-f001:**
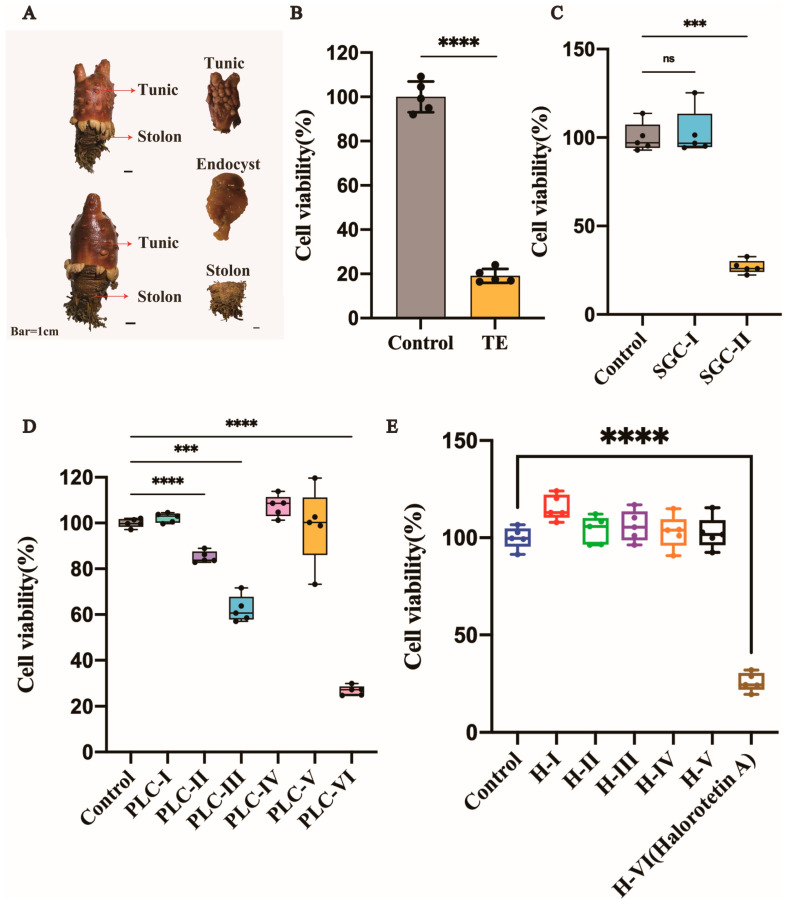
Isolation and activity verification of Halorotetin A. (**A**) *Halocynthia roretzi* and its three components. Bar = 1 cm (**B**) Relative cell viability of HepG-2 Cells after treatment with TE (200 μL/mL). Data represent the mean ± s.e.m.; *n* = 5 samples. Significance was determined by the *t*-test after Welch’s correction. (**C**–**E**), Relative cell viability of HepG-2 Cells after treatment with SGC-I~II (600 μg/mL) (**C**) and PLC-I~VI (50 μg/mL) (**D)** and H-I~VI (25 μg/mL) (**E**). Significance was determined by the one-way ANOVA test after Brown–Forsyths and Welch test. For boxplot: The vertical line represents 1.5 IQR, the upper and lower hinges present the 75% quantiles and 25% quantiles, respectively, and the center line means the median value. Each point represents a sample; *** *p* < 0.001, **** *p* < 0.00001.

**Figure 2 marinedrugs-21-00051-f002:**
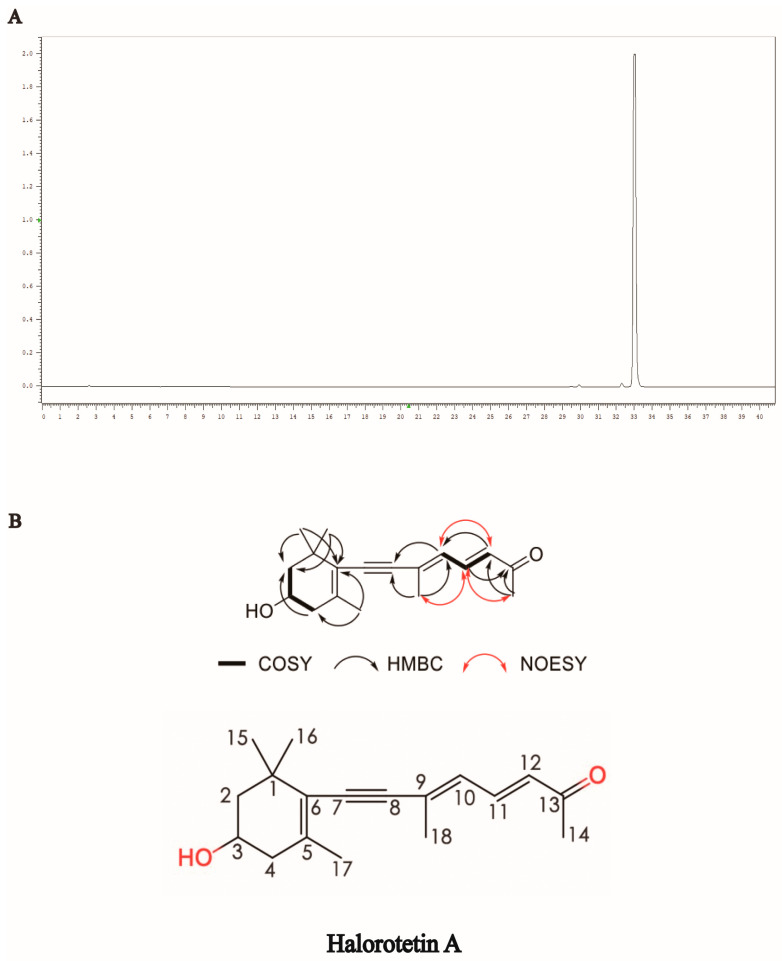
Structural identification of Halorotetin A. (**A**) The purity detection of Halorotetin A for subsequent structural identification. (**B**) The key COSY, HMBC, and NOESY correlations and chemical structure of Halorotetin A.

**Figure 3 marinedrugs-21-00051-f003:**
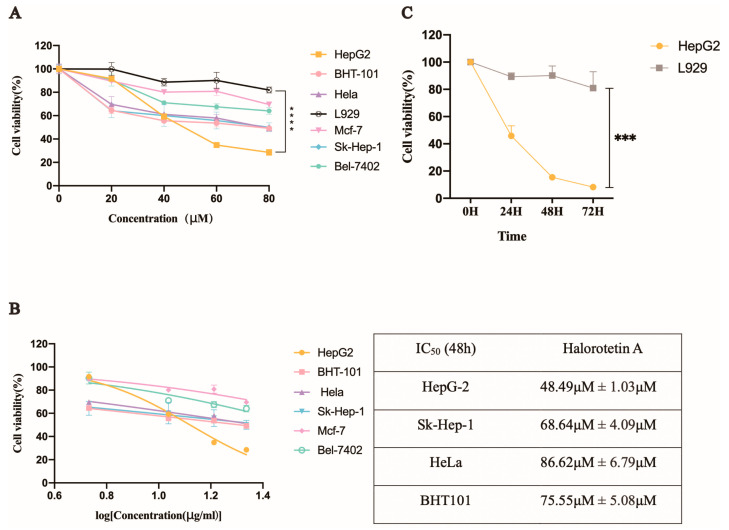
The inhibitory effects of Halorotetin A and the IC_50_ values on different tumor cells. (**A**) Relative cell viability of different tumor cells after treatment with different Halorotetin A concentrations. (**B**) IC_50_ values of Halorotetin A on tumor cells. All curves were generated by the nonlinear regression. (**C**) Relative cell viability of HepG-2 cells and L929 after treatment with Halorotetin A at the concentration of 60 μM for 24, 48, and 72 h, respectively. Data represent the mean ± s.e.m.; *n* = 5 samples. Significance was determined by the one-way ANOVA test after Brown–Forsyths and Welch test. *** *p* < 0.001, **** *p* < 0.00001.

**Figure 4 marinedrugs-21-00051-f004:**
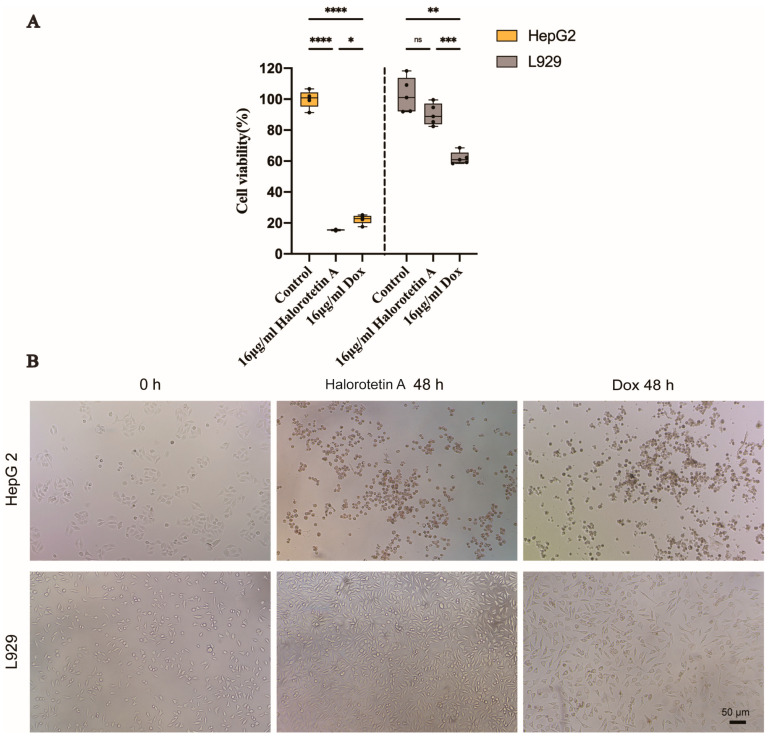
The cytotoxicity of Halorotetin A on HepG-2 cells and L929 cells. (**A**) Relative cell viability of HepG-2 and L929 cells treated with 16 μg/mL Halorotetin A and 16 μg/mL dox for 48 h, respectively. The vertical line represents 1.5 IQR, the upper and lower hinges present the 75% quantiles and 25% quantiles, respectively, and the center line means the median value. Each point represents a sample. Significance was determined by the one-way ANOVA test after Brown–Forsyths and Welch test. * *p* < 0.05, ** *p* < 0.01, *** *p* < 0.001, **** *p* < 0.00001 (**B**) Morphological features of L929 and HepG-2 cells treated with 16 μg/mL Halorotetin A and 16 μg/mL dox for 48 h, respectively, as observed under light microscopy. Bar = 50 μm.

**Figure 5 marinedrugs-21-00051-f005:**
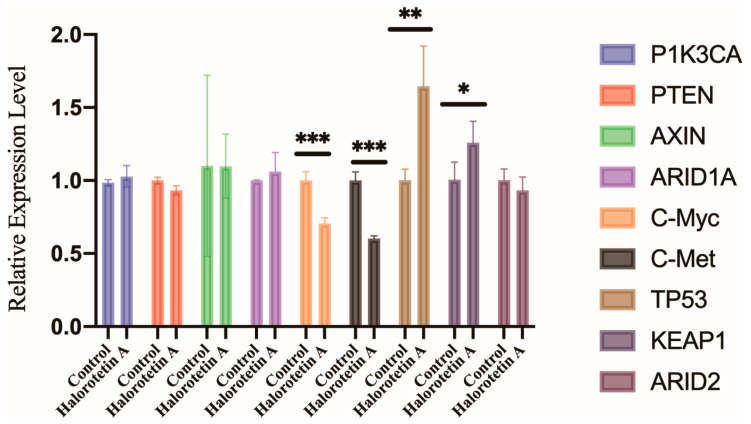
Gene expression profiles of HepG-2 cells after Halorotetin A treatment. qRT-PCR showed the expression levels of multiple oncogenes and suppressor genes in HepG-2 cells with or without 40 μM Halorotetin A treatment for 16 h. The same volume of methanol was used as the experimental group. Data represent the mean ± s.e.m.; *n* = 4 samples. Significance was determined by the t-test after Welch’s correction. * *p* < 0.05, ** *p* < 0.01, *** *p* < 0.001.

**Table 1 marinedrugs-21-00051-t001:** NMR spectral data of compound Halorotetin A in Methanol-*d_*4*._*

No.	δ_H_ (m, *J* in Hz)	δ_C_ (m)	HMBC
1		36.1	
2	1.42 (m) 1.81 ddd (12.4, 3.7, 2.0)	46.0	C-1,3,15,16
3	3.90 dddd (11.9, 9.3, 5.6, 3.6)	63.7	
4	2.03 (m) 2.41 (m)	40.8	C-2,3,5,6,17
5		139.8	
6		123.7	
7		92.2	
8		97.0	
9		130.0	
10	6.46 d (11.6)	131.6	C-,8,11,12,18
11	7.56 dd (15.3, 11.6)	138.6	C-12,13
12	6.21 d (15.3)	129.6	C-10,13,14
13		200.0	
14	2.30 (s)	26.1	C-12,13
15	1.13 (s)	27.7	C-1,2,6,16
16	1.17 (s)	29.7	C-1,2,6,15
17	1.91 (s)	21.3	C-4,5,6,7
18	2.11 (s)	17.3	C-8,9,10

## Data Availability

All data generated or analyzed during this study are included in the manuscript and supporting files.

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
