# Peer review of "Halorotetin A: A Novel Terpenoid Compound Isolated from Ascidian Halocynthia rotetzi Exhibits the Inhibition Activity on Tumor Cell Proliferation"

_marinedrugs, 2023, doi:10.3390/md21010051_

Round 1
Reviewer 1 Report
The manuscript entitled (A novel terpenoid compound isolated from ascidian Halocynthia rotetzi exhibits the inhibition activity on tumor cell proliferation) by Li et al. reported the isolation, purification, and structure elucidation of one novel from ascidian Halocynthia rotetzi. The structure of the compound was established by detailed analyses of the HRESIMS and NMR data. Liver carcinoma cell line (HepG-2 cells) was used for toxicity activity of the compound. The author said that the compound is a novel, but unfortunately, I did not have access to SciFinder check for that.
The manuscript should be revised well, and the following issues should be covered
1- The MS is poorly written there are different typing and grammatical mistakes that should be throughout the manuscript
2- Author said in discussion of compound H-V1 (The 1H and 13C NMRs, 1H-1H COSY, HSQC, HMBC, NOESY and HRESIMS were then applied to get the profile data). Dear authors, in the chemistry of natural product, especially when we talk about new compounds, it’s not acceptable that author said this. So please discuss the 1D and 2D NMR the compound intensively. Also, IR should be included in the discussion. The discussion should be more informative and convincing. This is a novel compound and need more discussion and more proves to elucidate the compound.
3- A trivial name should be given to the new compound. The title should be modified accordingly. Also, the chemical name of the compound should add.
4- Dear author (The 1H and 13C NMRs) is not corrected so the correct is (The 1H and 13C NMR).
5- How is the stereochemistry of OH group at C-3 assigned?.
6- How authors identified Halocynthia rotetzi? Please give all information about collection, identification, and voucher specimen code.
7- Figure 1 part A is unreadable please increase resolution and font size.
8- Also, resolution of figure 4 should be improved.
9- In extraction and isolation of the compound, author should add percentage of solvent system used in Colum Chromatography in details, also Rt of the HPLC.
10- Abbreviation should be added and written in full name when it is firstly mentioned.
11- What is the nature of the compound, it is oily, crystalline, …….and what is quantity of the compound. How the purity of the compound was identified.
12- The NMR data should be written in table.
13- The IR and m.p. (if crystalline) should be reported and added in the discussion.
14- Concerning to the MTT assay, what is the positive control used and its value, also the tested concentrations of compound and control used in this assay.
15- The biosynthetic pathways of this compound should be added and discussed.
Author Response
Reviewer 1:
The manuscript entitled (A novel terpenoid compound isolated from ascidian Halocynthia rotetzi exhibits the inhibition activity on tumor cell proliferation) by Li et al. reported the isolation, purification, and structure elucidation of one novel from ascidian Halocynthia rotetzi. The structure of the compound was established by detailed analyses of the HRESIMS and NMR data. Liver carcinoma cell line (HepG-2 cells) was used for toxicity activity of the compound. The author said that the compound is a novel, but unfortunately, I did not have access to SciFinder check for that.
Response: Through the access to SciFinder of OUC, the search results of the compound were shown in the following figures.
The MS is poorly written there are different typing and grammatical mistakes that should be throughout the manuscript.
Response: We have read through the manuscript and corrected the typo and grammatical mistakes.
Author said in discussion of compound H-V1 (The 1H and 13C NMRs, 1H-1H COSY, HSQC, HMBC, NOESY and HRESIMS were then applied to get the profile data). Dear authors, in the chemistry of natural product, especially when we talk about new compounds, it’s not acceptable that author said this. So please discuss the 1D and 2D NMR the compound intensively. Also, IR should be included in the discussion. The discussion should be more informative and convincing. This is a novel compound and need more discussion and more proves to elucidate the compound.
Response: We thank reviewer for the suggestion. The detailed process of analyzing the structure of the compound has been shown in Result 2.2 in the revision.
A trivial name should be given to the new compound. The title should be modified accordingly. Also, the chemical name of the compound should add.
Response: Thanks for the suggestion. The compound was named (DAT-1), and the chemical name of the compound was (3E,5E)-8-(4-hydroxy-2,6,6-trimethylcyclohex-1-enyl)-6-methylocta-3,5-dien-7-yn-2-one. The information was added in Result 2.2. in the revision.
Dear author (The 1H and 13C NMRs) is not corrected so the correct is (The 1H and 13C NMR).
Response: Thanks for the suggestion. We have made changes in the manuscript.
How is the stereochemistry of OH group at C-3 assigned?
Response: Thanks for the comments. The stereochemistry of OH group at C-3’ was R-configuration through the polarization detection.
How authors identified Halocynthia rotetzi? Please give all information about collection, identification, and voucher specimen code.
Response: The Halocynthia roretzi species was originally introduced from Korea into China. The samples used in this study were collected from the XUNSHAN group (Weihai, Shandong, China), where large scale aquaculture has been realized.
Figure 1 part A is unreadable please increase resolution and font size.
Response: The resolution and font size were increased.
Also, resolution of figure 4 should be improved.
Response: Thanks for the suggestion. The low resolution may be due to the fact that the cells were in the cell culture plate when the pictures were taken, and the lid of the plate affected the resolution of the pictures.
In extraction and isolation of the compound, author should add percentage of solvent system used in Colum Chromatography in details, also Rt of the HPLC.
Response: Thanks for the suggestion. The relevant information has been added in methods 4.2 and 4.3.
Abbreviation should be added and written in full name when it is firstly mentioned.
Response: Thanks for the suggestion. The full name of PLC was added in result 2.1 when it was firstly mentioned.
What is the nature of the compound, it is oily, crystalline, …….and what is quantity of the compound? How the purity of the compound was identified.
Response: Thanks for the comments. The nature of the compound was the light-yellow solid. The quantity of the compound was approximately 7mg. The purity of the compound was identified by semipreparative-HPLC (Figure 2A).
The NMR data should be written in table.
Response: Thanks for the suggestion. The NMR data has been written in table in methods 4.4.
The IR and m.p. (if crystalline) should be reported and added in the discussion.
Response: Thanks for the suggestion. The discussion of IR was added in the result 2.2, and the IR spectrum was added in figure S9.
Concerning to the MTT assay, what is the positive control used and its value, also the tested concentrations of compound and control used in this assay.
Response: Thanks for the comments. The positive control was doxorubicin, the concentration of the dox was 20 μg/mL for each experiment. The concentrations of compound were in each figure legend. The control group was added with the same concentration of methanol as in the experimental group.
The biosynthetic pathways of this compound should be added and discussed.
Response: Thanks for the comments. Many bacteria have been identified from the gut of H. roretzi. The compound that we identified in this study could be produced by H. roretzi itself or by its symbiont. We discussed the source of the DAT-1 in discussion (line 515-519) of manuscript.

Reviewer 2 Report
Jianhui Li et al. identified a novel compound with anti-cancer activity from Halocynthia rotetzi. They tested several cancer lines and a mouse fibroblast cell line and found the compound is more sensitive in HepG-2. Then they examined the gene expression at the RNA level by RT-PCR. This manuscript has an interesting and complete story, but the anti-cancer study in this manuscript is a little simple. But it is fine for a natural product journal, hence this manuscript should put more words on the structure elucidation of this new compound. However, the NMR spectrum is not friendly to identify the exact structure. I would recommend as a major revision. Either authors could provide a purer and clearer NMR spectrum or define the exact chemical shifts that belong to the new compound and shifts of impurity will be acceptable.
Please address below concerns:
Line 83 PLC add the full name.
Figure 1B and C viability use % of control, and Y axis uses 100. Including the rest of the figure.
Line 103: Need more details of structure elucidation. From ESI mass to determine the formula, detailed HMBC, COSY description to confirm the structure, including the configuration of the alkene group. I would suggest trying to determine the absolute configuration at C-3 as well.
Figure S2: can’t see the signal clearly and seems to have a lot of impurities.
Figure S3: label the exact chemical shift on the figure.
Figure S6: HMBC a little bit messy.
Figure 2: need COSY HMBC labeled on the structure. I would suggest labeling the chemical shift of each H and C on the structure in the separate figure in SI.
Figure 3: the highest concentration of this new compound kills less than 50% of most tested cell lines, which indicated that need to increase the concentration, or this compound is not potent for its anti-cancer activity (IC50 just fine actually). Hence, I would not suggest highlighting this in text, just moderate activity.
Figure 4A: please be consistent with the same uM concentration unit. And it’s better just to determine the IC50 of DOX in HepG-2. When they all kill 90% nearly, it can’t use for comparison with only one concentration.
Figure 5: Please note that the description should be gene transcription level changes, not protein level which needs to be proved by WB.
Author Response
Reviewer 2:
Jianhui Li et al. identified a novel compound with anti-cancer activity from Halocynthia rotetzi. They tested several cancer lines and a mouse fibroblast cell line and found the compound is more sensitive in HepG-2. Then they examined the gene expression at the RNA level by RT-PCR. This manuscript has an interesting and complete story, but the anti-cancer study in this manuscript is a little simple. But it is fine for a natural product journal, hence this manuscript should put more words on the structure elucidation of this new compound. However, the NMR spectrum is not friendly to identify the exact structure. I would recommend as a major revision. Either authors could provide a purer and clearer NMR spectrum or define the exact chemical shifts that belong to the new compound and shifts of impurity will be acceptable.
Response: Thanks for the comments and suggestions. We have added the detailed process of analyzing the structure of the compound. But unfortunately, due to the equipment or other reasons, we cannot obtain higher purity compound for structural identification, and we will improve the separation and purification conditions to obtain better experimental results in the future.
Line 83 PLC add the full name.
Response: Thanks for the suggestion. The full name of PLC was added in result 2.1.
Figure 1B and C viability use % of control, and Y axis uses 100. Including the rest of the figure.
Response: Thanks for the suggestion. The figures were changed, and the cell viability was used % of control and Y axis was used 100.
Line 103: Need more details of structure elucidation. From ESI mass to determine the formula, detailed HMBC, COSY description to confirm the structure, including the configuration of the alkene group. I would suggest trying to determine the absolute configuration at C-3 as well.
Response: We thank reviewer for providing this information. The detailed process of analyzing the structure of the compound is shown in Result 2.2. And the absolute configuration of OH group at C-3’ was R-configuration through the polarization detection.
Figure S2: can’t see the signal clearly and seems to have a lot of impurities.
Response: Thanks for the comments. We have undergone multiple HPLC separations and purifications, possibly due to the column reached its limits and the compound have some impurities.
Figure S3: label the exact chemical shift on the figure.
Response: Thanks for the comments. We have labeled the exact chemical shift on the figure S3.
Figure S6: HMBC a little bit messy.
Response: Thanks for the comments. Due to the equipment or other reasons, we cannot obtain higher purity compound for structural identification, and we will improve the separation and purification conditions to obtain better experimental results in the future.
Figure 2: need COSY HMBC labeled on the structure. I would suggest labeling the chemical shift of each H and C on the structure in the separate figure in SI.
Response: Thanks for the suggestion. We have labeled the structure on COSY and HMBC.
Figure 3: the highest concentration of this new compound kills less than 50% of most tested cell lines, which indicated that need to increase the concentration, or this compound is not potent for its anti-cancer activity (IC50 just fine actually). Hence, I would not suggest highlighting this in text, just moderate activity.
Response: Thanks for the suggestion. We added ‘although the compound could inhibit the proliferation of different tumor cell lines, a higher concentration of action was required to achieve stronger cytotoxicity’ in result 2.3.
Figure 4A: please be consistent with the same uM concentration unit. And it’s better just to determine the IC50 of DOX in HepG-2. When they all kill 90% nearly, it can’t use for comparison with only one concentration.
Response: Thanks for the suggestion. We have exhibited the same concentration in the figure 4A. It cannot use for comparison with only one concentration; hence, we have revised the expression in result 2.4.
Figure 5: Please note that the description should be gene transcription level changes, not protein level which needs to be proved by WB.
Response: Thanks for the suggestion. We modified the text in result 2.5.
Round 2
Reviewer 1 Report
1- The name of compound should be derived from the name of Halocynthia rotetzi so, I suggested the following trivial name of the compound which is (Halorotetin A).
2- Title of the paper should change to: (Halorotetin A: a novel terpenoid isolated from ascidian Halocynthia rotetzi exhibits the inhibition activity on tumor cell proliferation).
3- Please replace all DAT-1 to Halorotetin A in all manuscript
4- The NMR table that in experimental part should move after discussion part and cite it in the discussion. Please draw and write your data in a table like this
Table 1. NMR spectral data of compound VS-2-1 (Oleanolic acid, CDCl3, 850 and 213 MHz)
|
No. |
dH (m, J in Hz) |
C (m)δ |
HMBC |
|
|
|
|
|
|
|
|
|
|
Like this table
5- In exponential part please add all other data for your compound as Physicall character, m.p. UV, IR, HRMS…… please try to show and read a paper contain NMR data in Marine Drug Journal.
6- Please add a conclusion and write it carefully supported by your data
Author Response
Reviewer 1:
The name of compound should be derived from the name of Halocynthia rotetzi so, I suggested the following trivial name of the compound which is (Halorotetin A).
Response: Thanks for the suggestion. We accepted the name of the compound.
Title of the paper should change to: (Halorotetin A: a novel terpenoid isolated from ascidian Halocynthia rotetzi exhibits the inhibition activity on tumor cell proliferation).
Response: Thanks for the suggestion. We are agree with the suggested title.
Please replace all DAT-1 to Halorotetin A in all manuscript.
Response: Yes, we have replaced it in the manuscript.
The NMR table that in experimental part should move after discussion part and cite it in the discussion. Please draw and write your data in a table like this.
Response: Thanks for the suggestion. We have drawn and wrote our NMR data in a table and added it in result 2.2.
In exponential part please add all other data for your compound as Physicall character, m.p. UV, IR, HRMS…… please try to show and read a paper contain NMR data in Marine Drug Journal.
Response: Thanks for the suggestions. The UV, IR and HRESIMS data were added in the result 2.2. The compound was not crystalline, so the m.p. was not detected.
Please add a conclusion and write it carefully supported by your data
Response: Thanks for the suggestion. We have added the conclusions in the manuscript.
Reviewer 2 Report
The authors addressed most of the concerns. However, I am still concerned about the NMR spectrum and elucidation. Please see the below suggestions:
1. Line 85 Preparative Layer Chromatography-PLC
2. Figure 3B, change to 100% as well, Y axis.
3. Line 177, change 60uM to ug/ml
4. Table of NMR data in Part 4.4, move into the result part as an individual table. MASS data move into the second sentence of the description part in 2.2
5. I didn’t see the COSY and HMBC labelled in the structure. If a table of NMR is put in the result, it’s ok to not label individual carbon or hydrogen shifts in structure. But 2D NMR signal must do, all COSY-Bold band, and HMBC-black arrows, NOESY-red arrow.
6. Line 123, I didn’t see the COSY of H-10/11/12 in the spectrum, only one COSY signal for two Hydrogens 7.21/7.56.
7. In NMR of H, 4.87 biggest peak of H2O in the sample, which may make H and COSY HMBC weak. I would suggest making the sample totally dry, then doing the H and COSY, HMBC again. Ideally, for one new natural product, this weak NMR with some impurity is not acceptable. I can understand that impurity always exists in the natural product isolation field, hence, at least it’s necessary to make signals stronger in intensity.
8. I didn’t see the text concluded for “OH group at C-3’ was R-configuration” should put the end of part 2.2.
9. Line 200 it’s "t-test". Not uppercase.
Author Response
Reviewer 2:
Line 85 Preparative Layer Chromatography-PLC
Response: Thanks for the suggestion. We have changed it in the manuscript.
Figure 3B, change to 100% as well, Y axis.
Response: Thanks for the suggestion. We have changed the figure 3B to 100% as well.
Line 177, change 60uM to ug/ml.
Response: Thanks for the suggestion. We have changed it in manuscript.
Table of NMR data in Part 4.4, move into the result part as an individual table. MASS data move into the second sentence of the description part in 2.2.
Response: Thanks for the suggestions. We have drawn and wrote our NMR data in result 2.2 and the MASS data was moved into the second sentence of the description part in 2.2.
I didn’t see the COSY and HMBC labelled in the structure. If a table of NMR is put in the result, it’s ok to not label individual carbon or hydrogen shifts in structure. But 2D NMR signal must do, all COSY-Bold band, and HMBC-black arrows, NOESY-red arrow.
Response: Thanks for the suggestions. The all COSY-Bold band, and HMBC-black arrows, NOESY-red arrows were all added in the manuscript (Figure 2B).
Line 123, I didn’t see the COSY of H-10/11/12 in the spectrum, only one COSY signal for two Hydrogens 7.21/7.56.
Response: Thanks for the comments. We may have missed some details before. Hence, we have made the signals stronger in COSY spectrum. The COSY of H-10/11/12 in the spectrum was more clear now in the revision.
In NMR of H, 4.87 biggest peak of H2O in the sample, which may make H and COSY HMBC weak. I would suggest making the sample totally dry, then doing the H and COSY, HMBC again. Ideally, for one new natural product, this weak NMR with some impurity is not acceptable. I can understand that impurity always exists in the natural product isolation field, hence, at least it’s necessary to make signals stronger in intensity.
Response: Thanks for the suggestions. We have made the signals stronger in all 2D spectrums and made the signals stronger in intensity.
I didn’t see the text concluded for “OH group at C-3’ was R-configuration” should put the end of part 2.2.
Response: Thanks for the suggestion. We have added it in the manuscript.
Line 200 it’s "t-test". Not uppercase.
Response: Thanks for the suggestion. We have changed it in manuscript.
Round 3
Reviewer 1 Report
In Table heading, the heading is not correct so,
Please correct (m, J in HZ)Hd) to dH (m, J in Hz) please note that the alphabets Z, H, and C....... Capital and subscript. Also, δ(m)c change to dC (m).
Like this table

Author Response
Please correct (m, J in HZ)Hd) to dH (m, J in Hz) please note that the alphabets Z, H, and C....... Capital and subscript. Also, δ(m)c change to dC (m).
Response: We have corrected the title of table 1 in the manuscript.
Reviewer 2 Report
The authors addressed most concerns. Please correct the below suggestions before publication.
1. Figure S2, better show the UV curve, if have.
2. All the CH3 CH2 should be subscript, please correct in the whole manuscript.
3. Line 134, I still didn’t see the data which supports “In addition, the 134 absolute configuration of OH group at C-3’ was R-configuration.” Please add some supporting results before this sentence. Otherwise, this couldn’t be confirmed as an absolute configuration.
4. Table 1 legends write the solvent for NMR info. and not right for each column title, please find and follow a published paper
5. All figures are low resolution, please fit the requirements of publishment in the journal.
Author Response
Reviewer 2:
Figure S2, better show the UV curve, if have.
Response: Thanks for the suggestion. We have shown the UV curve in the Figure S2. Due to our instrument, we could only detect the UV curve of Halorotetin A in the range of 210 nm to 390 nm.
All the CH3 CH2 should be subscript, please correct in the whole manuscript.
Response: We have corrected it in the revision.
Line 134, I still didn’t see the data which supports “In addition, the 134 absolute configuration of OH group at C-3’ was R-configuration.” Please add some supporting results before this sentence. Otherwise, this couldn’t be confirmed as an absolute configuration.
Response: Thanks for the suggestion. We have added it in the manuscript. The relevant references were listed below:
Thomas Andersson, Babak Borhan, Nina Berova, et, al. Absolute configurational assignment of 3-hydroxycarotenoids. J. Chem. Soc, 2000, 2409-2414.
Takao Matsuno, Masahiro Ookubo, Ichiro Shimizu, et, al. Carotenoids of sea squirts I new marine carotenoids, Halocynthiaxanthin and Mytiloxanthinone from Halocynthia roretzi. Chem. Pharm. Bull. 1984, 4309-4315.
Table 1 legends write the solvent for NMR info. and not right for each column title, please find and follow a published paper.
Response: Thanks for the suggestion. We have corrected the title of the table 1 in the manuscript.
All figures are low resolution, please fit the requirements of publishment in the journal.
Response: We have changed the high-resolution figures in the revision.